# Evolution Modes, Types, and Social-Ecological Drivers of Ecologically Critical Areas in the Sichuan–Yunnan Ecological Barrier in the Last 15 Years

**DOI:** 10.3390/ijerph19159206

**Published:** 2022-07-27

**Authors:** Xinyu Shi, Xiaoqing Zhao, Junwei Pu, Pei Huang, Zexian Gu, Yanjun Chen

**Affiliations:** 1School of Earth Sciences, Yunnan University, Kunming 650500, China; xinyushi@mail.ynu.edu.cn (X.S.); pujunwei@mail.ynu.edu.cn (J.P.); hphyyy09@mail.ynu.edu.cn (P.H.); guzexian@mail.ynu.edu.cn (Z.G.); iamcyj@mail.ynu.edu.cn (Y.C.); 2Institute of International Rivers & Eco-Security, Yunnan University, Kunming 650500, China; 3Forest Resource Management Division, Nujiang Forestry and Grassland Administration, Lushui 673100, China

**Keywords:** ecologically critical area, evolution modes and types, driving factors, spatiotemporal heterogeneity, Sichuan–Yunnan ecological barrier

## Abstract

The ecological barrier is a complex ecosystem that couples the human–nature relationship, and the ecologically critical area is an irreplaceable area with a special value in the ecosystem. Therefore, protecting the ecologically critical area is vital for maintaining and improving regional ecological security. Limited research has been conducted on the evolution of ecologically critical areas, and none of the studies have considered the spatiotemporal heterogeneity of the driving factors for different evolution modes and types. Therefore, this research adopts the ecologically critical index, landscape expansion index, and the random forest model to analyze the pattern, driving factors, and its spatial-temporal heterogeneity to the evolution modes and specific types of ecologically critical areas in the Sichuan–Yunnan ecological barrier area in the last 15 years. The results showed that: (1) the ecologically critical areas in the Sichuan–Yunnan ecological barrier have changed dramatically, with the area reduction being 61.06%. Additionally, the spatial distribution characteristics of the ecologically critical area from north to south include planar, point, and linear forms. (2) The evolution trend of the ecologically critical area is ‘degradation–expansion–degradation’. Spread is the predominant type of expansion mode, whereas atrophy is the predominant type of degradation mode, indicating that the evolution mainly occurs at the edge of the original ecologically critical areas. (3) In general, precipitation, area of forest, area of cropland, and GDP have contributed significantly to the evolution of ecologically critical areas. However, the same driving factor has different effects on the expansion and degradation of these areas. Expansion is driven by multiple factors at the same time but is mainly related to human activities and land use change, whereas for degradation, climate and policy are the main driving factors. The present research aimed to quantitatively identify the evolution modes and specific types of ecologically critical areas and explore the spatiotemporal heterogeneity of driving factors. The results can help decision-makers in formulating ecological protection policies according to local conditions and in maintaining and enhancing the regional ecological functions, thereby promoting the sustainable development of society-economy-ecology.

## 1. Introduction

Global climate change has been posing great risks to terrestrial ecosystems [1]. Additionally, the rapid socioeconomic development has caused a series of ecological problems. To safeguard national ecological security, China has built the ‘two barriers and three belts’ ecological security strategic pattern [2,3]. As a complex ecosystem that couples the human–nature relationship [4], an ecological barrier has good self-maintenance and internal regulatory capabilities, thus protecting the surrounding area or ensuring large-scale environmental security externally [5], which are fundamental to human development [6]. Under the background of global change, the ecological environment in the national ecological barrier has great spatiotemporal uncertainty [7], which has introduced great difficulties in the formulation of ecological policies and projects.

Ecologically critical area (ECA) is the important or fragile area with a particular value for the protection of ecosystem functions or biodiversity and resource productivity [8]. With rapid urbanization and industrialization, the degradation of the ecological environment is becoming increasingly serious [9]. ECA is highly sensitive to interference by humans. The degeneration or disappearance of these areas has a serious impact on the stability of the regional ecosystem [10,11]. An ecological barrier is a multiplexed system composed of multiple ecological projects with regional resources, the environment, and the economy as the background. This barrier not only considers the overall planning of the whole region and all factors but also coordinates the priority of construction and protection [12,13]. Therefore, identifying ECA from the geographical perspective [14] is of great significance for ensuring ecological safety.

As early as 1973, the United States termed the areas with fragile ecological environments as sensitive to external disturbance and having an irreplaceable role in maintaining biodiversity as the “critical area” [15]. In 1984, the Conservation Foundation formally proposed the concept of ECA, arguing that ECA should receive special attention and protection as they serve unique functions compared with other areas [16]. However, under limited economic and social resources, protecting all biodiversity areas is challenging [17]. Myers [18] proposed the hotspot concept on the basis of the research on the degree of threat to tropical rain forests. Subsequently, in 2000, Myers [19] proposed 25 priority hotspots for global biodiversity conservation. In the Global 200 Project sponsored by the World Wide Fund for Nature (WWF) [20], the list of global biodiversity priority conservation areas was constructed based on the theory of biodiversity conservation and the ecological regions [21], and the global biodiversity priority protection areas were divided into 233 ecological regions according to the main habitat types [22].

Currently, a general method for quantitatively identifying the spatial features of ECA evolution trends is lacking. Although many scholars have constructed a landscape expansion index to express the dynamic spatiotemporal pattern of urban expansion [23,24,25], only a few studies have applied it to ecological protection. Previous studies on the driving mechanisms of ECA evolution have focused on landscape pattern changes and ecological function changes [26] but lacked an assessment of the spatial and temporal heterogeneity of the driving factors. Related studies have revealed some common factors such as topography, accessibility, climate, and socioeconomic [27,28]. Studying these factors can contribute to the development of relevant ecological policies to protect ECA [29].

The Sichuan–Yunnan ecological barrier (SYEB) is an important part of the ‘two barriers and three belts’ ecological security strategic pattern, which plays an irreplaceable role in improving the ecological quality of key ecological function areas and enhancing ecosystem service capabilities. However, existing studies of the ecological barrier lack quantitative study, with a particular shortage of research on ECA evolution and its driving factors.

To fill the aforementioned research gap, the present research explored the spatiotemporal heterogeneity of ECA evolution and its driving factors at the county scale, considering the SYEB in China. The objectives of this research are as follows: (1) mapping the area of ECA expansion and degradation at the county scale at three time intervals (2005–2010, 2010–2015, and 2015–2019); (2) defining the specific types of ECA expansion mode and degradation mode during 2005–2010, 2010–2015, and 2015–2019 by using the landscape expansion index; and (3) quantifying the spatiotemporal heterogeneity between cropland evolution and its driving factors through the random forest model from the environmental, socioeconomic, and policy perspectives.

## 2. Materials and Methods

### 2.1. Study Area

The SYEB is part of the Sichuan–Yunnan-loess plateau ecological barrier and is a key component of China’s ecological security strategic patterns. The SYEB is located in the Hengduan Mountains and the transition zone from the Yunnan–Guizhou Plateau to the Qinghai–Tibet Plateau, wherein the topography is decreasing from northwest to southeast and the ecological environment is relatively fragile (Figure 1). This region is dominated by plateau mountain temperate and subtropical monsoon climate, with annual precipitation of 500–1400 mm, and the average annual temperature in the region can vary by more than 20 °C [30]. The SYEB has a special ecological status and plays an important role in soil conservation, water conservation, and biodiversity maintenance.

### 2.2. Data and Processing

Table 1 shows the names, formats, and sources of the data used in this research. The boundaries were extracted from the China National Ecological Barrier Area dataset, which is produced on the basis of a combination of Google Earth satellite images and vector boundary maps of Chinese counties, etc. According to the land classification system and our research problem, we grouped the data into seven categories, namely cropland, forest, grassland, water, wet land, impervious, and other land. The slope data were generated based on the DEM data. We integrated all data into a 1 km × 1 km grid for a unified analysis.

### 2.3. Methodology

#### 2.3.1. Research Framework

In this research, we evaluated the ecological significance of each grid and identified the ECA, classified the evolution modes and specific types, and explored its driving factors at the county scale. The overall research framework is shown in Figure 2 and it comprised five steps: (1) calculating the ecological critical index (ECI) of each grid during 2000, 2005, 2010, 2015, and 2019 through the county-scale spatial analysis by comprehensively considering the ecosystem service function and landscape ecological security; (2) determining the threshold value and identifying ECA based on the frequency distribution characteristics of ECI; (3) determining the expansion mode and degradation mode of ECA evolution during 2005–2010, 2010–2015, and 2015–2019 and classifying the specific types of expansion mode and degradation mode by landscape expansion index (LEI); (4) selecting the explanatory variables according to the research topic and previous research and then putting the dependent variable and the explanatory variables into the random forest (RF) model and running the model; and (5) analyzing the spatiotemporal heterogeneity of the driving factors for evolution modes and specific types.

#### 2.3.2. Ecological Critical Index

By coupling the ecosystem service function and landscape ecological security, the ECI was constructed to represent the degree of ecological importance at the county scale. ECI is the basis for identifying ECA and is calculated using the following formula:(1)$${\mathrm{ECI}}_{i}=\sqrt{{\mathrm{MESLI}}_{i}\times {\mathrm{LESSI}}_{i}}$$
where ${\mathrm{ECI}}_{I}$ is the ecological critical index of the ith grid; ${\mathrm{MESLI}}_{I}$ is the multiple ecosystem services landscape index of the ith grid; and ${\mathrm{LESSI}}_{I}$ is landscape ecological structure security index of the ith grid.

MESLI can be used to effectively identify the ability of a region to provide multiple ecosystem services at the same time [31], which is a comprehensive and valuable environmental indicator for identifying cold and hot spots for multiple ecosystem services [32]. It is calculated using the following formula:(2)$$\mathrm{MESLI}=\sum_{i=1}^{n}\frac{x_{i}-\min(x_{i})}{\max(x_{i})-\min(x_{i})}$$
where i is the type of ecosystem service function; n is the number of ecosystem service function; and x_i_, max(x_i_), and min(x_i_) are the observed value, maximum value, and minimum value of the ith ecosystem service function, respectively.

According to the functional orientation of the SYEB mentioned in the “*Master plan for major projects for the protection and restoration of important ecosystems nationwide*”, four types of ecosystem services are selected, namely carbon sequestration, water production, soil conservation, and habitat quality. Referring to previous studies, NPP was used to represent the carbon sequestration function, and the water production services, soil conservation, and habitat quality were calculated using the INVEST model [33,34].

LESSI evaluates the security of the landscape structure from three perspectives, namely landscape vulnerability, boundary fragmentation, and landscape type fragmentation [35], and it has been widely used in the landscape ecological security assessment [36,37]. LESSI is calculated using the following formula:LESSI = 1 − [(PD + ED) × 2.5V](3)
V = α × AWMSI + β × F + γ × D(4)
where PD is the patch density in the landscape class, ED is the edge density in the landscape class, and V is the landscape fragmentation. AWMSI is the area-weighted mean shape index; F is fractal dimension; D is division; and α, β, and γ are the weights of AWMSI, F, and D, respectively, which are assigned as 0.5, 0.3, and 0.2, respectively.

#### 2.3.3. Identifying the Evolution Modes and Specific Types of ECA

Traditional landscape indices can only reflect the static spatial distribution of the landscape [38], whereas LEI can quantitatively describe the dynamic change process [39]. The formula for calculating LEI is as follows:(5)$$\mathrm{LEI}=100\times \frac{A_{0}}{A_{E}-A_{P}}$$
where A_0_ is the area of the original ECA grid in the buffer zone; A_E_ is the area of buffer zone; and A_P_ is the area of new ECA; the range of LEI values is between 0 and 100. Considering the area of the studied region, we set the buffer radius as 1 km. All ECA grids whose buffers overlap belong to the same patch and have the same LEI.

Through the spatial analysis, we divided the ECA evolution into two modes: expansion and degradation. Grids that were previously identified as non-ECA and later identified as ECA are called expansion. By contrast, grids that were previously identified as ECA and later identified as non-ECA are called degradation (Figure 3).

According to the definition of LEI [39], we have defined six types of ECA evolution. The expansion mode has three specific types: (1) if the buffer of a new ECA patch consists of non-ECA only (LEI = 0), it is classified as the isolation type, implying that the new ECA patch is spatially discontinuous with the original ECA patch; (2) if the buffer of a new ECA patch consists of non-ECA mixed with original ECA (0 < LEI ≤ 50), it is classified as the spread type, meaning that the new ECA patch is located at the edge of the original ECA patch; and (3) if the buffer of a new ECA patch is mostly occupied by the original ECA (50 < LEI ≤ 100), it is classified as the infilling type, indicating that the new ECA patch is located within the original ECA patch. The degradation mode also has of three specific types, namely departed, atrophy, and disintegration. Their classification rules correspond to the specific type of expansion mode.

#### 2.3.4. Random Forest Model

The RF model is a data mining method for categorical regression trees proposed by Breiman [40]. RF is an effective classifier that is composed of a set of tree-structured classifiers {h (x, θk), k = 1, ……}, where {θk} represents explanatory, identically-distributed random vectors. With the input of the explanatory variable X, each DT casts a unit vote for the most popular class [41].

In this research, we chose the minimum Gini value method as the segmentation criterion. The minimum Gini value of an internal tree node was calculated using the following formula:(6)$$\mathrm{Gini}\left(t\right)=1-\sum_{q=1}^{u}{\left[p\left(q|t\right)\right]}^{2}$$
where p(q|t) represents the probability of the risk class q at node t; u represents the number of classes.

In this study, the decrease in the Gini index at the node split was used to calculate the importance of each index to the result of risk classification. The formula used for calculation is as follows:(7)$$P\left(r\right)=\frac{\sum_{i=1}^{k}\sum_{j=1}^{t}D_{\mathrm{Grij}}}{\sum_{r=1}^{m}\sum_{i=1}^{k}\sum_{j=1}^{t}D_{\mathrm{Grij}}}\times 100\%$$
where m represents the total number of indices, k is the number of texturing trees, t is the number of nodes in each tree, D_Grij_ is the Gini decrease value at the jth node in the ith tree that belongs to the rth index, and Pr is the degree of contribution of the rth index from all available indices.

#### 2.3.5. Determination of Explanatory Variables

According to the previous research and the theoretical analysis, the potential driving factors that could influence the ECA evolution were grouped into two categories: environment condition and socioeconomic development (Table 2).

DEM (x1), slope (x2), temperature (x3), precipitation (x4), area of forest (x5), and area of grassland (x6) were selected as the six explanatory variables to represent environmental conditions from the perspectives of terrain, meteorology, and ecosystem composition. The DEM and slope determine whether the conditions of a region are suitable for ecosystem succession to higher levels. Precipitation and temperature determine ecosystem succession by affecting vegetation growth. Land use reflects the regional ecosystem composition, and we selected the major land use components in the ECA including forest and grassland.

Socioeconomic factors indicate the impact of human activities, with the most important factors being the area of cropland (x7), GDP (x8), population (x9), and nightlight (x10). These four explanatory variables are direct representations of human activity. We chose the distance from impervious (x11) and cropland (x12) to represent ECA location, reflecting the ease with which ECA is affected by these regions. This research could be improved by considering these factors, thus providing a better understanding of ECA evolution.

## 3. Results

### 3.1. Spatial-Temporal Pattern of ECA

By counting and plotting the frequency distribution of ECI during 2005, 2010, 2015, and 2019, we observed that the frequency distribution of ECI in each period is similar. By comparing the extraction results of different thresholds, including peak, trough, and inflection point, we finally chose the last peak of the trend line as the extraction threshold (T) of ECA (Figure 4). Therefore, the grids with ECI > T were identified as ECA, whereas those with ECI < T were identified as non-ECA. 

The last 15 years have witnessed a serious decline in the number of ECA grids, from 49,068 to 19,105, representing an overall decrease of 61.06%. ECA had the largest number of grids, reaching 49,068, in 2005. In 2010, the number of ECA grids reduced to 33,300. In 2015, the number of ECA grids increased slightly to 41,169. Finally, in 2019, the number of ECA grids drastically reduced to 19,105.

According to the aforementioned identification process, we determined the spatial distribution of the ECA of the SYEBZ in 2005, 2010, 2015, and 2019. The results showed a significantly heterogeneous distribution of ECA (Figure 5). The ECAs in the northern part of the study area were distributed in a planar form and located mainly in Mianyang, Deyang, Chengdu, Ya’an, Meishan, and Leshan. In the central part, the ECAs were distributed in a pointed form, and located in Ganzi, Liangshan, and Panzhihua. The ECAs in the southern part were distributed in a linear form and located mainly in Lijiang, Nujiang, and Dali.

### 3.2. Spatial-Temporal Distribution of Evolution Trends of ECA

#### 3.2.1. Evolution Modes of ECA

The ECA evolution trend was “degradation–expansion–degradation”. From 2005 to 2010, the number of expanded grids was 1537 and the number of degradation grids was 17,305; the direction of ECA evolution mainly showed a degradation trend. The expansion grids were mainly located at the junction of Chengdu, Leshan, and Ya’an, whereas the degradation grids were concentrated in Mianyang, Deyang, Chengdu, and Meishan in the north. From 2010 to 2015, the number of expansion grids was 11,248, the number of degradation grids was 3379, and the direction of ECA evolution mainly showed an expansion trend. The expansion grids were mainly concentrated in Mianyang, Aba, Chengdu, Ya’an, and Meishan in the northern part of the region. The distribution of the degradation grids was more discrete, located mainly in the south of the region. From 2015 to 2019, the number of expansion grids was 2271, the number of degradation grids was 24,335, and the ECA evolution showed mainly the degradation trend. The expansion grids showed mainly a discrete distribution in the central part of the region, and the degradation grids were mainly concentrated in Mianyang, Aba, Ya’an, Meishan, and Leshan in the northern region (Table 3, Figure 6).

#### 3.2.2. Specific Types of Evolution Modes

Among the three specific types of the expansion mode, the predominant type is spread (Figure 7), indicating that expansion mainly starts at the edge of original ECA. In the last 15 years, as for the isolation type, the proportion of grid count increased from 5.70% to 10.45%, and the proportion of patch area increased from 10.30% to 16.75%, indicating that the spatial distribution of the isolation type tends to be concentrated. For the spread type, the proportion of grid count increased from 70.14% to 79.58%, and the proportion of patch area increased from 54.20% to 65.23%, indicating that the spatial distribution of the spread type shifted from concentrated to dispersed. Finally, for the infilling type, the proportion of grid count decreased from 24.16% to 9.97%, and the proportion of patch area decreased from 35.50% to 18.02%, showing that the dispersed characteristic is more obvious for this type.

Atrophy is the predominant type among the three specific types of the degradation mode (Figure 7), indicating that degradation also mainly starts at the edge of the original ECA. In the last 15 years, for the departed type, the proportion of grid count increased from 4.92% to 5.77%, and the proportion of patch area increased from 22.54% to 29.74%, indicating that the spatial distribution of departed type tends to be more dispersed. For the atrophy type, the proportion of grid count increased from 86.09% to 92.47%, and the proportion of patch area decreased from 57.21% to 52.05%, indicating that the spatial distribution of the atrophy type tends to be more concentrated. Finally, for the disintegration type, the proportion of grid count decreased from 9.00% to 1.76%, and the proportion of patch area decreased from 20.25% to 18.21%, indicating that the distribution of this type tends to be more dispersed.

### 3.3. Spatial-Temporal Heterogeneity of Driving Factors

#### 3.3.1. Verification of RF Model Accuracy

RF models were run with two modes of ECA evolution, three specific types of expansion modes, and three specific types of degradation modes as predictor variables, respectively. The percentage of correct classification results was used to evaluate the accuracy of the RF model. According to Table 4, the highest prediction accuracy was 97.96%, whereas the lowest prediction accuracy was 68.86%, and the average accuracy reached 83.98%. These data suggest that the results of the RF model are credible.

#### 3.3.2. Heterogeneity of Driving Factors to the Evolution Modes and Specific Types of ECA

The importance of explanatory variables to ECA evolution was classified from low to high as follows: I (0–0.5037%), II (0.5038–1.5381%), III (1.5382–3.7400%), IV (3.7401–8.4272%), V (8.4273–18.4052%), and VI (18.4053–39.6460%) (Figure 8) by the geometric interval grading method. 

The environment is the primary driving factor. The importance level of DEM (x1) was relatively stable, with it being IV for all three periods. The importance level of slope (x2) showed a decreasing trend, exhibiting levels V, III, and II during 2005–2010, 2010–2015, and 2015–2019, respectively, suggesting that with the gradual adaptation of the ecosystem to the terrain, the impact of the terrain gradually decreases. The importance of temperature (x3) also showed a decreasing trend, with the level being IV in 2005–2010 and decreasing to III in 2010–2015 and 2015–2019. The importance level of precipitation (x4) showed a fluctuating upward trend, with it being V in 2005–2010, decreasing to IV during 2010–2015, and then increasing to VI during 2015–2019. In the context of climate change, hydrothermal conditions are crucial for ECA evolution. The importance level of area of forest (x5) was V during 2005–2010 and 2010–2015, which decreased to II during 2015–2019. The importance level of area of grassland (x6) increased from II in 2005–2010 to IV in 2010–2015 and finally restored to II in 2015–2019, indicating that the importance of natural ecosystem background conditions for ECA is gradually decreasing.

In terms of socioeconomic factors, the importance level of area of cropland (x7) was IV in 2005–2010, which increased to VI during 2010–2015 and 2015–2019, indicating that the impact of artificial ecosystems is gradually increasing. The importance level of GDP (x8) was stable, with it being V in all three periods. The importance level of population (x9) decreased from IV in 2005–2010 to II in 2010–2015 and 2015–2019. The importance level of nightlight (x10) also showed a downward trend, decreasing from level III in 2005–2010 to level I during 2010–2015 and 2015–2019, indicating the importance of the economic output. Owing to urbanization, rural population and economic activities are gradually being concentrated in cities; hence, the importance of human activity gradually decreased. The importance level of distance from impervious (x11) showed a fluctuating trend, increasing from IV in 2005–2010 to V in 2010–2015 and then decreasing to III in 2015–2019. The importance of distance from cropland (x12) has increased from level II in 2005–2010 to level III in 2010–2015 and 2015–2019. The Chinese government has proposed a ‘new urbanization’ policy to alleviate the ecological pressure caused by rapid urbanization; therefore, the impact of urban expansion first increased and then decreased. In addition, due to the implementation of the Cropland Balance Policy, cropland will gradually move out with urban expansion.

To explore the differences in the effects of each variable between ECA expansion and degradation, we used three expansion types or three degradation types as predictor variables. As shown in Figure 8, the length of the rectangle reflects relative importance. The solid rectangles and dashed rectangles represent the relative importance of the explanatory variables for three specific types of expansion mode and three specific types of degradation mode, respectively. During 2005–2010, the explanatory variables with greater relative importance to the type of expansion were nightlight (x10) and distance from impervious (x11), whereas the explanatory variable with greater relative importance to the type of degradation was precipitation (x4). During 2010–2015, the explanatory variables with greater relative importance to the type of expansion were area of forest (x5) and area of cropland (x7), whereas the explanatory variables with greater relative importance to the type of degradation were area of grassland (x6) and distance from impervious (x12). During 2015–2019, the explanatory variables with greater relative importance to the type of expansion were area of forest (x5), area of cropland (x7), nightlight (x10), and distance from impervious (x12), whereas the explanatory variables with greater relative importance to the type of degradation were temperature (x3) and precipitation (x4). Overall, forest ecosystems have abundant species and a complex network structure with good stability. Thus, large areas of forest are conducive to ECA expansion. As an artificial ecosystem, the agroecosystem is closely and intricately linked to the socioeconomic sphere of human beings. Under the Cropland Balance Policy, economic development and urban expansion have contributed to the outward expansion of cropland, leading to a negative impact on ECA expansion. However, for ECA degradation, climate factors including temperature and precipitation are more significant. Climate change is aggravating the imbalance of water and heat distribution and thus promoting ECA degradation.

## 4. Discussion

### 4.1. Characteristics of the ECA Evolution Modes and Types

The spatial distribution characteristics of ECA from north to south include planar, point, and linear forms. In the northern part of the study area, ECA is mainly distributed in the marginal mountain areas of the Sichuan Basin. The warm and humid climate has contributed to the increased diversity of animal and plant resources in the area; therefore, the spatial pattern of ECA is planar. The central part of the study area is the middle section of the Hengduan Mountains, where the soil is barren, vegetation is sparse, and the ecological environment is fragile. It is also difficult to form a concentrated and contiguous habitat patch there, so the spatial pattern of ECA is a point pattern. In the southern part of the study area, the Jinsha, Lancang, and Nu rivers flow in parallel, wherein the biological communities are enriched, and the ECA is distributed linearly along the sides of the valley.

ECA has undergone dramatic changes in the last 15 years, and its evolutionary trend is degradation–expansion–degradation. The socioeconomic development owing to rapid urbanization inevitably causes different degrees of damage to the ecosystem [42], thereby causing ECA degradation. To alleviate the ecological pressures of rapid development, China has implemented various extensive ecological projects to protect and expand its forests [43]. Central and Southwest China is one of the key regions for afforestation, and the forest cover has increased significantly in a short time period, which is the main reason for the ECA expansion in the study area in 2015. However, due to the low stability of single species afforestation, forest plantation is more dependent on human management [44]. Therefore, planted forest ecosystems are more susceptible to degradation than natural ecosystems.

The competition between the expansion and degradation processes is most intense at the edge of the ECA, which often represents the junction or overlapping area between different ecosystems. This is the transition zone of the community where species penetrate each other and is the area with obvious environmental gradient changes [45]. Therefore, ECA has a profound edge effect. The edge effect interpenetrates climate, vegetation, and landscape in the area and undergoes gradient mutations, leading to increased environmental heterogeneity [46] and making ECA edges vulnerable to degradation. However, due to edge effects, ecological engineering can enhance the overall ecological benefits. Therefore, the government should implement different policies to promote expansion and curb degradation according to the spatial distribution characteristics and evolution pattern of ECA, which can help in enhancing regional ecological functions and maintaining regional ecological security.

### 4.2. Understanding the Spatiotemporal Heterogeneity of Driving Factors

Overall, the explanatory variables identified as having great importance in ECA evolution are precipitation (x4), area of forest (x5), area of cropland (x7), and GDP (x8). Precipitation is an important factor in ecosystem succession, and climate change has been causing significant damage and increasingly irreversible losses to the ecosystem [47]. To relieve ecological pressure, China has implemented various ecological projects such as afforestation and reforestation projects since the 20th century [48]. In the “*Outline of National Ecological Fragile Area Protection Plan*”, the SYEB mainly belongs to the southwest mountainous ecologically fragile area with interlocking agriculture and animal husbandry, where the future development direction needs to strictly return farmland to forests and close hillsides for afforestation. Especially in the “*National Main Functional Area Planning*”, it is further clarified that the SYEB needs to focus on strengthening the functions of soil erosion control and natural vegetation protection, which have contributed to the greening of China [49,50]. GDP represents the economic output of a region and is the primary representation of the intensity of human activity [51]. Therefore, these explanatory variables have strong impacts on ECA evolution.

The explanatory variables identified as having the least effect on ECA evolution are population (x9) and nightlight (x10). This is because the ECA is located far from the city center and thus is sparsely populated and dimly lit. Additionally, the new urbanization policy [52] promoted by the Chinese government requires economic intensification and ecological livability; therefore, the economic output of the region no longer depends only on the size of the population, and economic development is gradually becoming green and sustainable.

### 4.3. Policy Implication

Scientific ecological policies can promote the sustainable development of the region. Therefore, we propose site-specific policies to protect ECA based on the spatiotemporal pattern of ECA, spatiotemporal heterogeneity of driving factors to ECA evolution, and differences in the drivers of expansion and degradation.

The edges of ECA represent the intersection between natural and artificial ecosystems and are the areas where anthropogenic disturbances compete most fiercely with natural ecological restoration; hence, expansion and degradation occur mainly at the edges of ECAs. These areas require the construction of ecological isolation zones, delineation of ecological red lines, and other measures to control the intensity of interference due to human activities. The afforestation and reforestation projects have contributed significantly to increasing the forest area in the study area, which had a positive effect on ECA expansion. However, the importance of human management in this area cannot be ignored [50]. Therefore, the continuous management and monitoring of ecological projects are necessary to maintain the ecological functions of planted forests.

Specifically, the northern part of the SYEBZ is the Sichuan Basin, which has the highest population density and the strongest economic activity. The ECA in this region has a planar distribution and is heavily degraded. We suggest that in the future, urban development boundaries should be strictly controlled, and high-standard cropland construction should be implemented to alleviate the ecological pressure of urban and cropland expansions. Coordinating the contradiction between economic development and ecological protection is essential to curb ECA degradation. The central part of the SYEBZ is the middle section of the Hengduan Mountains, which is an ecologically fragile area with fewer human activities and a high degree of preservation of the original ecological environment. The ECA in this region has a point distribution. To promote ECA expansion in the region, a system of nature reserves should be built to improve ecological containment functions and the local climate. The southern part of SYEBZ, wherein three rivers flow in parallel, has a fragile ecological environment and poor socioeconomic development. The ECA in this region has a linear distribution. In this region, the concept of ecological priority and green development should be strictly implemented in the future. With ecological connotation as the premise, the advantages of water and heat conditions in the high mountain valley should be utilized, and ecological utilization-based agriculture should be vigorously developed.

## 5. Conclusions

This study constructs ECI to identify ECAs in the SYEBZ during 2005, 2010, 2015, and 2019. Then, the LEI was used to delineate the specific types of ECA expansion and degradation modes during three periods, namely 2005–2010, 2010–2015, and 2015–2019. Finally, the RF model was used to explore the spatiotemporal heterogeneity of drivers of ECA evolution modes and specific types.

The overall trend of ECA evolution is degradation–expansion–degradation. Among the three expansion types, namely isolation, spread, and infilling, the predominant type is spread. Similarly, among the three degeneration types, namely departed, atrophy, and disintegration, the predominant type is atrophy. This indicates that the evolution of ECA mainly occurred at the edge of the original ECA. The drivers having a great importance in ECA evolution are mainly precipitation (x4), area of forest (x5), area of cropland (x7), and GDP (x8), indicating that ECA expansion is driven by various factors but is primarily associated with land use change and anthropogenic factors. However, for ECA degradation, climate factors including temperature and precipitation are more significant. These conclusions are vital to the protection and management of the national ecological barrier zone and the promotion of coordinated regional socio-economic-ecological development.

## Figures and Tables

**Figure 1 ijerph-19-09206-f001:**
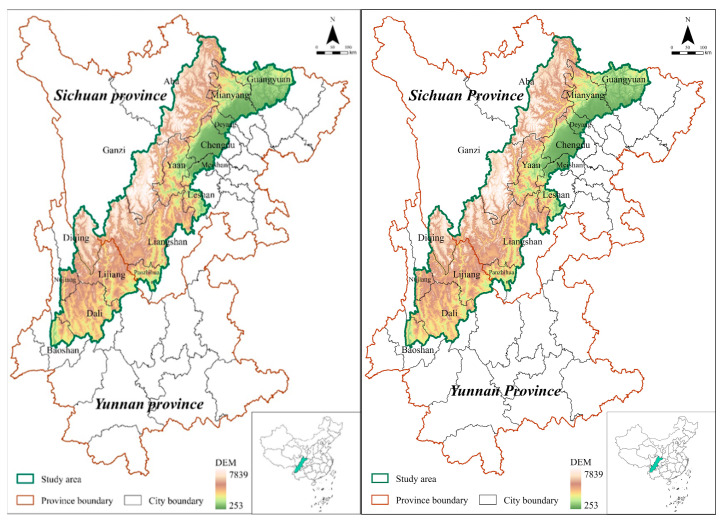
Study area.

**Figure 2 ijerph-19-09206-f002:**
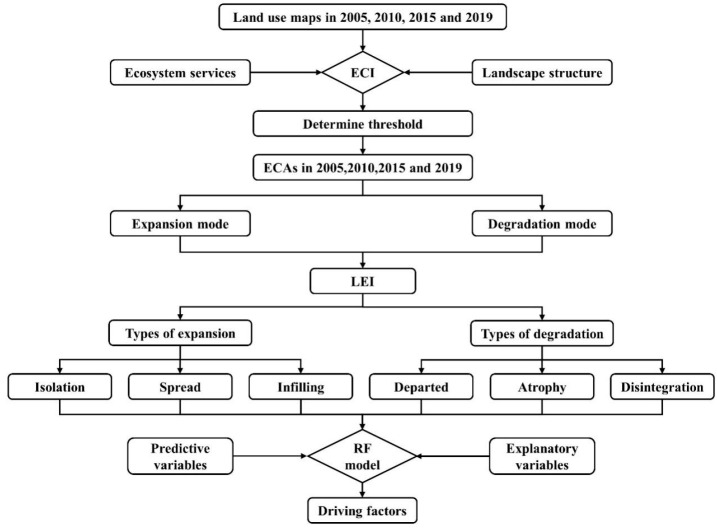
Research framework.

**Figure 3 ijerph-19-09206-f003:**
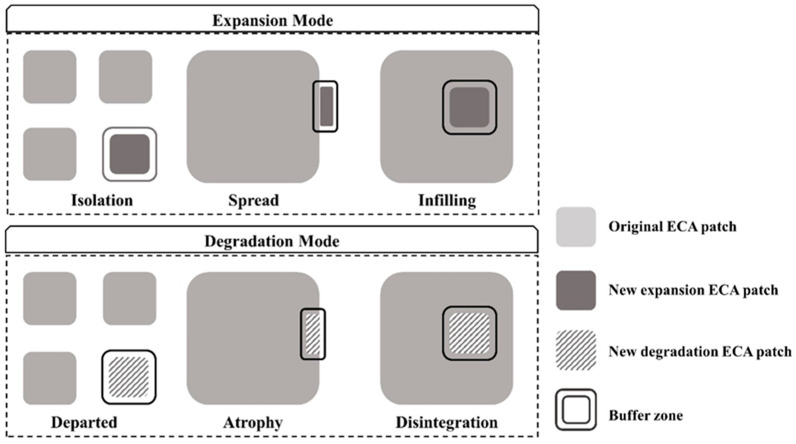
Graphical representation of evolution modes and specific types of ECA.

**Figure 4 ijerph-19-09206-f004:**
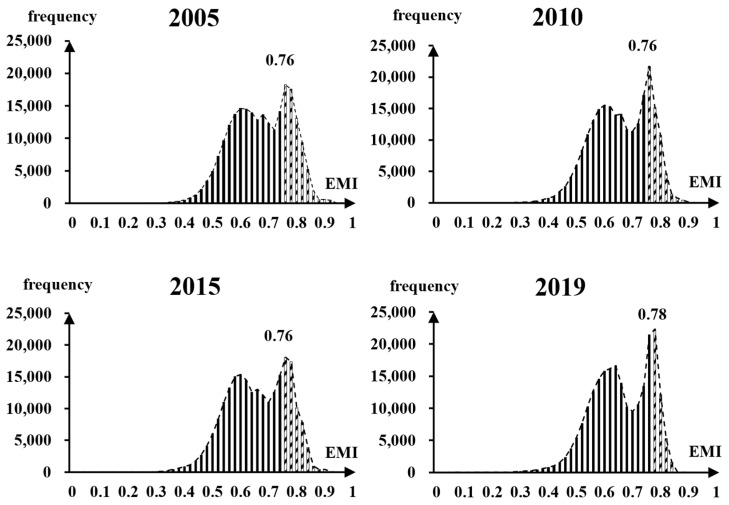
Frequency distribution histogram and extraction threshold of ECI.

**Figure 5 ijerph-19-09206-f005:**
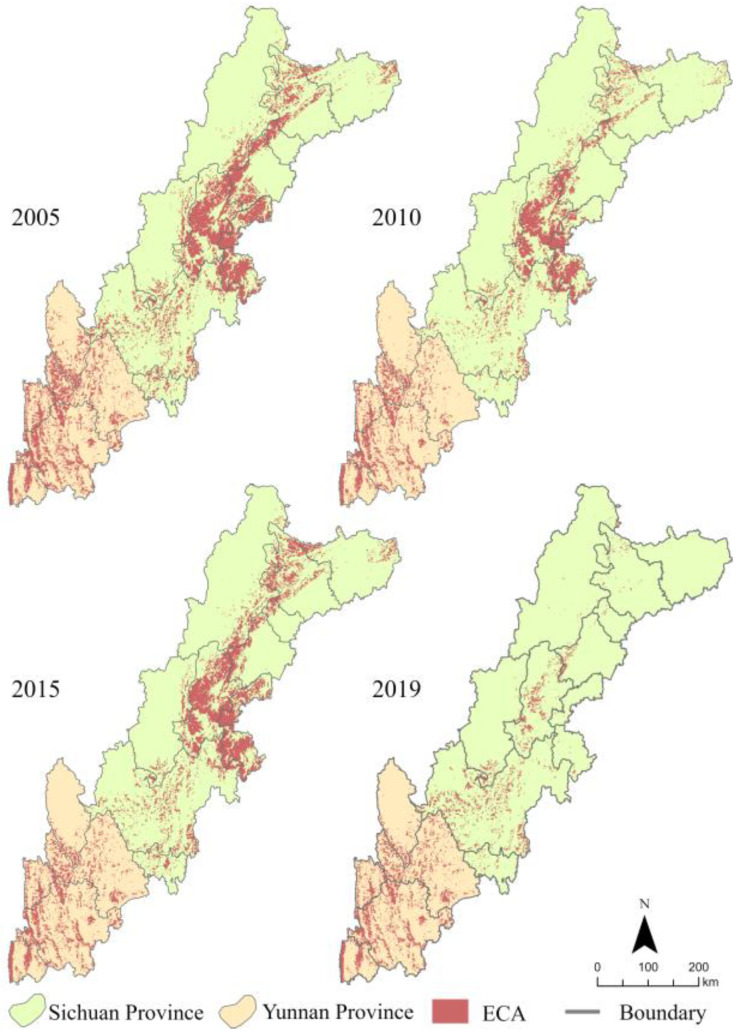
Spatial-temporal distribution of ECA.

**Figure 6 ijerph-19-09206-f006:**
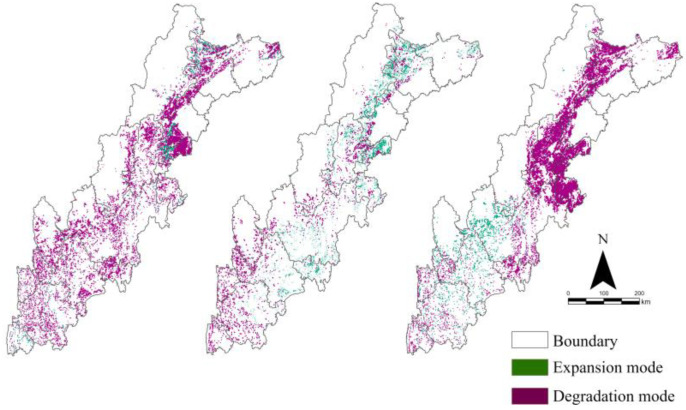
Spatial-temporal distribution of evolution modes.

**Figure 7 ijerph-19-09206-f007:**
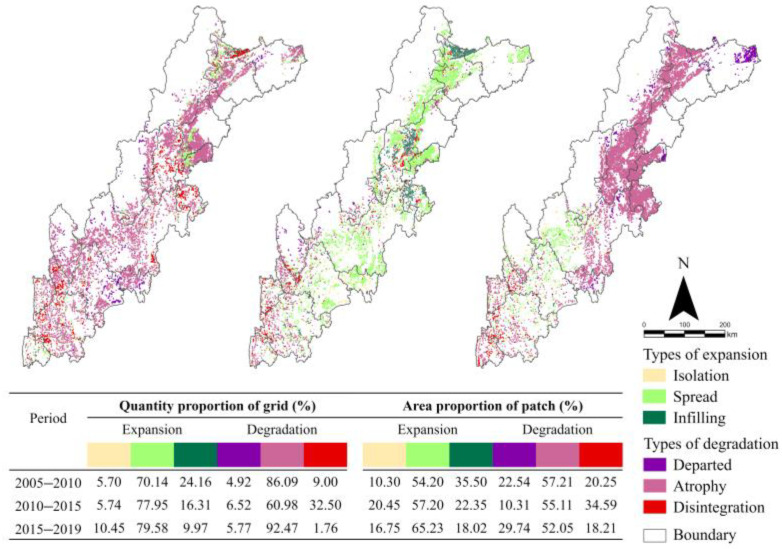
Spatial-temporal distribution of specific types and its quantity and area proportions.

**Figure 8 ijerph-19-09206-f008:**
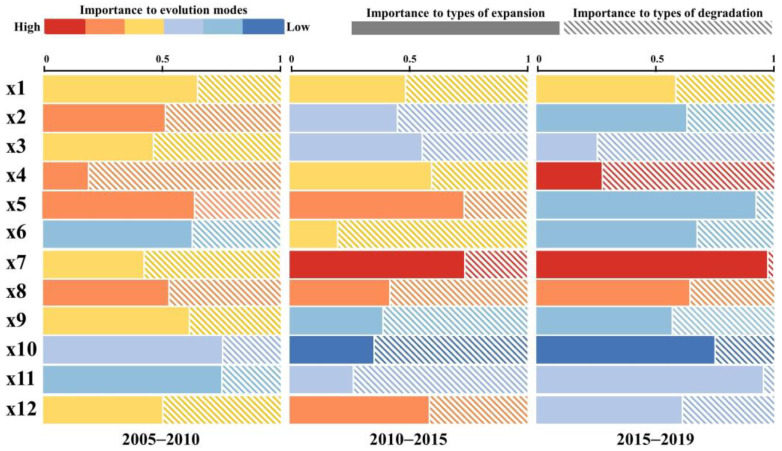
Variable importance of driving variables to evolution modes and specific types.

**Table 1 ijerph-19-09206-t001:** Data name, format, and source.

Data Name	Data Format	Resolution	Data Source
BarrierZoneChina.	shp	/	http://geodoi.ac.cn/ (accessed on 15 March 2021)
Land cover data	tif	30 m	https://zenodo.org/record/5210928 (accessed on 30 March 2021)
Digital terrain data	tif	30 m	http://www.gscloud.cn/ (accessed on 4 April 2021)
Meteorological data	nc	1 km	http://www.geodata.cn/ (accessed on 6 April 2021)
Soil data	mdb	1 km	https://www.fao.org/ (accessed on 6 April 2021)
NPP data	tif	500 m	https://ladsweb.modaps.eosdis.nasa.gov/ (accessed on 6 April 2021)
Nightlight data	tif	1 km	http://data.tpdc.ac.cn/ (accessed on 6 April 2021)
Population count	tif	100 m	https://www.worldpop.org/ (accessed on 8 April 2021)
GDP	tif	1 km	http://www.resdc.cn/ (accessed on 10 April 2021)

**Table 2 ijerph-19-09206-t002:** Factors and explanatory variables of driving factors for ECA evolution.

Factor	Variable	Description	Unit
Environmental condition	x1	Average altitude	m
	x2	Average slope	°
	x3	Average temperature	°C
	x4	Total precipitation	mm
	x5	Proportion of forest area	km^2^
	x6	Proportion of grassland area	km^2^
Socioeconomic development	x7	Proportion of cropland area	km^2^
	x8	Average GDP	million CNY/km^2^
	x9	Population density	number of people/km^2^
	x10_0_	Average night light	/
	x11	Euclidean distance from impervious	km
	x12	Euclidean distance from cropland	km

**Table 3 ijerph-19-09206-t003:** Count and proportion of evolution modes.

Period	Evolution Modes			
	Expansion Mode		Degradation Mode	
	Count	Proportion	Count	Proportion
2005–2010	1537	8.16%	17305	91.84%
2010–2015	11248	76.90%	3379	23.10%
2015–2019	2271	8.54%	24335	91.46%

**Table 4 ijerph-19-09206-t004:** Prediction accuracy of the random forest model.

Period	Evolution Modes	Types of Expansion Mode	Types of Degradation Modes
2005–2010	92.86%	72.50%	86.08%
2010–2015	82.19%	79.01%	68.86%
2015–2019	97.96%	82.83%	94.75%
Statistics	Max	Min	Mean
	97.96%	68.86%	83.98%

## Data Availability

Not applicable.

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
