# Peer review of "Evolution Modes, Types, and Social-Ecological Drivers of Ecologically Critical Areas in the Sichuan–Yunnan Ecological Barrier in the Last 15 Years"

_ijerph, 2022, doi:10.3390/ijerph19159206_

Round 1
Reviewer 1 Report
Review: Evolution modes, types, and social-ecological drivers of ecologically critical areas in the Sichuan–Yunnan ecological barrier in recent 15 years
This paper discusses the extraction of ecologically critical areas, evolutionary patterns and the heterogeneity of their drivers. The manuscript is currently relatively complete in terms of scientific questions, research methods, research data, analysis of results, discussion and conclusions, and the paper is of good quality. After conducting minor revisions, it can be considered for publication.
Introduction section: the typicality and representativeness of the study area could be highlighted a bit more.
Materials and methods section: How were the boundaries of the study area determined needs to be given? The specification of the formula and its variables needs to be noted, and the method of determining ECI thresholds needs to be given in the methods section.
Results section: The values need to be clearly visible in the graph.
Discussion section: The results obtained in this paper need to be discussed on the basis of the actual situation in the study area, using the findings of previous studies.
All concerns, major and minor, are noted below.
1.Line22 - abbreviations need to have full names the first time they appear, ECA?
2. Line123 - the abbreviation needs to have the full name the first time it appears, LEI?
3. The formula needs further editing.
4. How the threshold value of ECI is determined, this is not explained in the method section.
5. How is the segmentation threshold of LEI determined?
6. The range of LEI values and their ecological significance need to be explained.
7. In section 2.3.5, how were the data, especially the population and GDP data, treated? The authors do not explain whether the boundaries of the study area are the administrative boundaries of the region, and if not, how were these data taken and what clarification is needed?
8. How did the authors consider the different thresholds for ECI to classify ECAs? The thresholds for 2005, 2010 and 2015 are 0.76, while the threshold for 2019 is 0.78. Are the ECA areas classified with different thresholds comparable?
9. In section 3.2.2, the values in the analysis of the results are not visible in the graph and need to be adjusted or a table added to the graph.
10. Can section 4.2 be discussed in the context of local realities and the results of this paper, such as how ecological conservation projects implemented by the government, etc., affect heterogeneity?
11. The format of the references needs to be carefully modified according to the requirements of the journal.
Author Response
Dear reviewer:
Thank you for your review comments! We have organized a response letter, please see the attachment.
Kind regards,
Xinyu Shi

Reviewer 2 Report
Dear Authors
your manuscript is interesting and of significance to technical communities.
Although it is well-organized, there are some gaps wherever you come to the section of methodology in general and the Ecological critical index. The readers of your manuscript get lost when he/she read about the components of your ECI formula.
To me, it is not clear how you have identified ecosystem services and why you have chosen 3- 4 ecosystem services out of a huge number of ecosystem services. How and why you have directedly applied an experimentally developed formula for landscape security? do not regional ecological feature affact the landscape security?
In overal, this part of your methodology might throughly be re-written to increase clearance of this section.
Author Response

(The authors gave the same response as above.)
